# NEURAL EULERIAN SCENE FLOW FIELDS

**Kyle Vedder**[1,2*]   **Neehar Peri**[2,3]   **Ishan Khatri**[3]   **Siyi Li**[1]   **Eric Eaton**[1]
**Mehmet Kocamaz**[2]   **Yue Wang**[2]   **Zhiding Yu**[2]   **Deva Ramanan**[3]   **Joachim Pehserl**[2]
[1]University of Pennsylvania    [2]NVIDIA    [3]Carnegie Mellon University

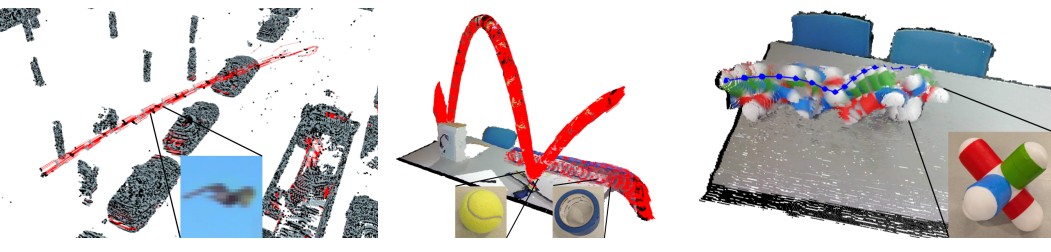

(a) Small object motion extraction...      (b) ...in diverse, dynamic scenes...      (c) ...with emergent 3D point tracking behavior!

Figure 1: EulerFlow is able to capture the motion of small, fast moving objects with few lidar points, such a bird flying in front of an autonomous vehicle (Figure 1a). EulerFlow's flexibility allows it to estimate scene flow for fast-moving table top objects *without additional hyperparameter tuning* (Figure 1b). EulerFlow's ODE estimate exhibits emergent 3D point tracking behavior without explicit long-horizon supervision (Figure 1c). Note that point clouds are shown in color for visualization purposes only; RGB is not used during optimization.

## ABSTRACT

We reframe scene flow as the task of estimating a continuous space-time ordinary differential equation (ODE) that describes motion for an entire observation sequence, represented with a neural prior. Our method, *EulerFlow*, optimizes this neural prior estimate against several multi-observation reconstruction objectives, enabling high quality scene flow estimation via self-supervision on real-world data. EulerFlow works out-of-the-box without tuning across multiple domains, including large-scale autonomous driving scenes and dynamic tabletop settings. Remarkably, EulerFlow produces high quality flow estimates on small, fast moving objects like birds and tennis balls, and exhibits emergent 3D point tracking behavior by solving its estimated ODE over long-time horizons. On the Argoverse 2 2024 Scene Flow Challenge, EulerFlow outperforms *all* prior art, surpassing the next-best *unsupervised* method by more than $2.5\times$, and even exceeding the next-best *supervised* method by over 10%. See vedder.io/eulerflow for interactive visuals.

## 1   INTRODUCTION

Scene flow estimation is the task of describing motion with per-point 3D motion vectors between temporally successive point clouds (Dewan et al., 2016; Liu et al., 2019; Erçelik et al., 2022; Jund et al., 2021; Zhang et al., 2024b; Vedder et al., 2024; Khatri et al., 2024). Such per-point motion estimates are critical for autonomy in diverse environments, e.g., maneuvering around open-world objects like debris (Peri et al., 2022a) or folding deformable cloth (Weng et al., 2022). Importantly, scene flow estimation requires not only an understanding of object *geometry*, but also its *motion*. However, scene flow methods broadly do not work on smaller objects (Khatri et al., 2024). For example, in the autonomous vehicles domain, Khatri et al. highlight that even supervised methods struggle to describe the majority of pedestrian motion, with unsupervised methods failing dramatically. Scene flow promises to be a powerful primitive for understanding the dynamic world, but such failures explain why it has limited adoption in downstream applications like tracking (Zhai et al., 2020) or open-world object extraction (Najibi et al., 2022).

---

*Corresponding email: kvedder@seas.upenn.edu

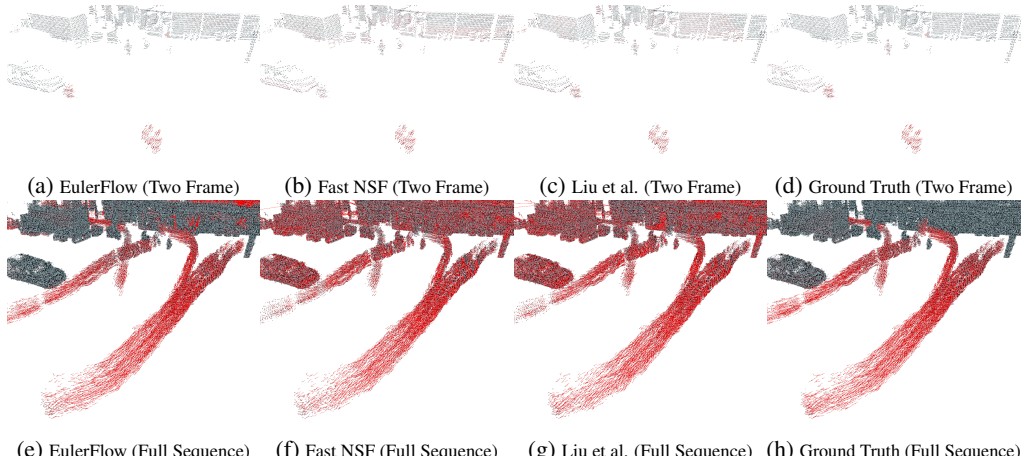

(a) EulerFlow (Two Frame)   (b) Fast NSF (Two Frame)   (c) Liu et al. (Two Frame)   (d) Ground Truth (Two Frame)

(e) EulerFlow (Full Sequence)   (f) Fast NSF (Full Sequence)   (g) Liu et al. (Full Sequence)   (h) Ground Truth (Full Sequence)

Figure 2: We visualize an example of five pedestrians crossing the street in front of a stopped car, cherrypicked to have unusually high density lidar returns, making it particularly easy to estimate flow. Figures 2a–2d depict a two-frame flow visualization of EulerFlow and several strong baselines. Notably, only visualizing flow over two frames makes it difficult to distinguish flow quality. In contrast, Figures 2e–2h depict flow vectors over the full sequence, making differences in quality clear; for example, EulerFlow is the only one without artifacts on the stopped car.

**Scene Flow via ODE.** In Figure 2, visual assessment of scene flow quality is much easier in an accumulated global frame; while incomplete due to an implicit time axis, these accumulated flow vectors allow viewers to imagine how positions in 3D space evolve over *many* timesteps, and compare that to predicted flows. This imagination of scene flow as continuous motion over large time intervals motivates us to model scene flow as an ordinary differential equation (ODE) that describes the scene's instantaneous motion across continuous position and time. Scene flow estimation then becomes the task of estimating this ODE. We can straightfowardly represent this ODE estimate with a neural prior (Li et al., 2021b) and optimize it against scene flow surrogate objectives, both over single frame pairs and extended *across arbitrary time intervals* to produce better quality estimates. We formalize this in Section 3 and propose the *Scene Flow via ODE* framework.

**EulerFlow.** We instantiate Scene Flow via ODE with standard point cloud distance objectives like Chamfer Distance to create *EulerFlow*. Notably, EulerFlow outperforms *all* prior art, supervised or unsupervised, on the Argoverse 2 2024 Scene Flow Challenge and Waymo Open Scene Flow benchmark. It outperforms prior *unsupervised* methods by a large margin ($> 2.5\times$ mean dynamic error reduction), and is able to capture small, fast moving objects, including those outside of labeled taxonomies (e.g. the flying bird in Figure 1a). Due to its simplicity, EulerFlow is able to provide high quality scene flow out-of-the-box on real-world data for other important domains such as dynamic tabletop settings (Figure 1b) *without* domain-specific tuning. Finally, we show that simple ODE solving techniques like Euler integration can be used to extract 3D point tracks (Figure 1c), which serves as both an exciting emergent behavior as well as a mechanism for visualizing and interpreting the quality of the continuous ODE estimate.

We present four primary contributions:

- We propose *Scene Flow via ODE* (SFvODE), a reframing of scene flow estimation as the task of fitting an ODE over all observations to describe the change of continuous positions over continuous time, unlocking a new class of surrogate objectives that enable better scene flow estimates.
- We instantiate SFvODE with *EulerFlow*, a flexible **unsupervised** scene flow method that achieves **state-of-the-art** performance on the Argoverse 2 2024 Scene Flow Challenge, **beating all prior supervised and unsupervised methods**.
- We study EulerFlow and show its strong performance is derived from its ability to optimize its ODE estimate against the full sequence of observations over arbitrary time horizons.
- We show that EulerFlow's simple, flexible formulation allows it to run unmodified on a variety of domains, with emergent capabilities like 3D point tracking behavior.

## 2 BACKGROUND AND RELATED WORK

**Evaluation.** Dewan et al. formalized scene flow for point clouds as the task of estimating motion between point cloud $P_t$ at time $t$ and point cloud $P_{t+1}$ at $t+1$ by estimating the true flow $\mathcal{F}_{t,t+1}$, i.e. true residual vectors for every point in $P_t$ that describe its movement to its associated position at $t+1$. Error is computed by measuring the per-point endpoint distance between estimated and ground truth vectors. Historically, these errors are reported with a per-point average (*Average EPE*); however, as Chodosh et al. show, Average EPE is dominated by background points, preventing meaningful measurement of non-ego object motion descriptions. Khatri et al. address this shortcoming with *Bucket Normalized EPE*, which reports per-class performance normalized by speed, allowing for direct comparisons across classes with very different average speeds (e.g. pedestrians and cars). Bucket Normalized EPE is the basis for the *Argoverse 2024 Scene Flow Challenge*[1], where methods are ranked by the mean error of their motion descriptions (*mean Dynamic Normalized EPE*).

**Input / Output Formulation.** Dewan et al.'s choice to formulate scene flow using *only* two input frames is arbitrary; it's the minimal information needed to extract rigid motion, but there are not real-world problems constrained to *only* have access to two frames. Indeed, using five or ten frames of past observations is standard practice in the 3D detection literature (Zhu et al., 2019; Vedder & Eaton, 2022; Peri et al., 2022b; 2023; Nalty et al., 2022), and multi-frame formulations have started to appear in the scene flow literature: Liu et al. (2024) and Flow4D (Kim et al., 2024) use three $(P_{t-1}, P_t, P_{t+1})$ and five input frames $(P_{t-3}, \ldots, P_{t+1})$ respectively to predict $\hat{\mathcal{F}}_{t,t+1}$. As we discuss in Section 3, rather than just using more observations to estimate flow for a single frame pair, we formulate scene flow as a joint estimation problem: given the full observation sequence $(P_0, \ldots, P_N)$, we estimate *all* flows $\hat{\mathcal{F}}_{0,1}, \ldots, \hat{\mathcal{F}}_{N-1,N}$ between *all* adjacent observations.

**Feedforward Methods.** Feedforward networks are a popular class of scene flow methods due to their fast inference speed (Liu et al., 2019; Behl et al., 2019; Tishchenko et al., 2020; Kittenplon et al., 2021; Wu et al., 2020; Puy et al., 2020; Li et al., 2021a; Jund et al., 2021; Gu et al., 2019; Battrawy et al., 2022; Wang et al., 2022b; Kim et al., 2024; Zhang et al., 2024a). While they are often trained with supervised labels, recent work has developed distillation pipelines that leverage unsupervised pseudolabelers (Vedder et al., 2024; Zhang et al., 2024b; Lin & Caesar, 2024).

**Neural Scene Flow Prior.** Li et al. (2021b) propose Neural Scene Flow Prior (NSFP), a widely adopted unsupervised scene flow approach. NSFP uses the inductive bias of the smooth, restricted learnable function class of two ReLU MLP coordinate networks (8 hidden layers of 128 neurons); $\theta$ to estimate forward flow and $\theta'$ to estimate backwards flow, minimizing

$$\text{TruncatedChamfer}(P_t + \theta(P_t), P_{t+1}) + \left\| P_t + \theta(P_t) + \theta'(P_t + \theta(P_t)) - P_t \right\|_2 \quad, \tag{1}$$

where TruncatedChamfer is defined as the standard $L_2$ Chamfer distance, but with per-point distances above 2 meters set to zero in order to reduce the influence of outliers. NSFP is optimized for at most 1000 steps with early stopping.

**Motion Beyond Two Frames.** Wang et al. (2022a) tackles the adjacent problem of estimating 3D point *trajectories* over 25 frames with Neural Trajectory Prior (NTP) by jointly optimizing three separate ReLU MLP neural priors: 1) a sinusoidal embedded, time conditioned, 25 frame trajectory basis estimator (embed$(t) \mapsto 256 \times 25 \times 3$ tensor, where 256 is the dimension of the trajectory basis), 2) a coordinate network bottleneck encoder, and 3) a bottleneck decoder to estimate a per-point linear combination over the learned trajectories. Trajectories are optimized for both a one-frame lookahead $L_2$ Chamfer loss and a cyclic consistency loss over the entire velocity space trajectory.

**Deformation in Reconstruction.** Nerfies (Park et al., 2021) and DynamicFusion (Newcombe et al., 2015) estimate a deformation field to warp a canonical frame to explain the observed frame. While capable of describing small motions, these methods require a canonical frame that contains all of the relevant geometry to deform; however, in large, highly dynamic scenes like autonomous driving, there is often no frame that contains all moving objects. By comparison, Scene Flow via ODE does not assume the existence of a canonical frame, instead only describing how the scene changes.

---

[1]https://www.argoverse.org/sceneflow

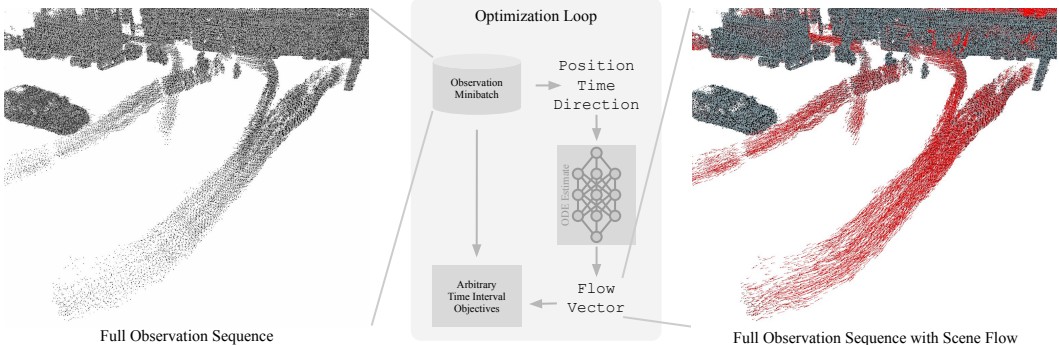

Figure 3: Overview of our *Scene Flow via ODE* framework, which estimates an ODE across the entire observation sequence by optimizing against multi-frame objectives. This ODE estimate is represented with a neural prior (Li et al., 2021b), providing a flexible, general representation for describing position-time motion.

## 3    SCENE FLOW VIA ODE

Prior art consumes multiple frames $(P_{t-N}, \ldots, P_{t+1})$ as input, but these methods are ultimately only tasked with estimating flow vectors between $P_t$ and $P_{t+1}$. We instead pose the problem of estimating a time-conditioned flow field that describes motion for *all* adjacent point clouds $P_t, P_{t+1}$ in the entire sequence $(P_0, \ldots, P_N)$. To do this, rather than describing scene flow as positional change over a fixed interval ($\mathcal{F}_{t,t+1}$ are residual vectors over the interval $t$ to $t + 1$) as we did in Section 2, we can instead express these changes as a differential equation that describes positional change over *continuous* time.

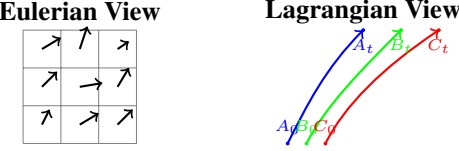

Figure 4: Comparison of Eulerian and Lagrangian descriptions of 2D flow. An Eulerian view characterizes a flow field via instantaneous velocities at many different points, while a Lagrangian view characterizes a flow field via trajectories of many different particles across time. Both approaches are valid ways of describing an underlying flow field, and with sufficient characterization one view can be readily converted to another, but the Lagrangian view relies on a definition of the definition of consistent canonical frame.

Formally, given a scene, let $L(x_0, y_0, z_0, t)$ be the Lagrangian view of the scene's true flow field, i.e. a continuous function that, given a canonical particle $(x_0, y_0, z_0)$ in a canonical frame 0, describes its $(x, y, z)$ position at frame $t$. As we discuss in Section 2, this Lagrangian view is common in the the deformable reconstruction literature, and the requirement for a canonical frame means such approaches struggle to describe scenes where there is no frame that contains all moving objects.

To break this canonical frame dependence, we choose to take an Eulerian view of motion, i.e. $F(x, y, z, t) = \left( \frac{\partial L_x}{\partial t}, \frac{\partial L_y}{\partial t}, \frac{\partial L_z}{\partial t} \right)$, which describes the velocity at position $(x, y, z)$ at time $t$. As we show in our derivation in Appendix C.1, this formulation does not require an external canonical frame to estimate a point's trajectory from $t$ to $t'$; instead, we can simply set the initial conditions of the ODE at $t$ to $x_t, y_t, z_t$ and utilize an off-the-shelf ODE solver (e.g. Euler integration) to extract flow from $t$ to $t'$, expressed as $E(x_t, y_t, z_t, t, t')$.

We do not know the true flow field $F$ when estimating scene flow; however, we can represent $F$ with a neural prior $\theta$ ($F \approx \theta$), and optimize $\theta$ against surrogate objectives. This framing, which we formalize into the *Scene Flow via ODE* framework (SFvODE; Figure 3), allows $\theta$ to benefit from constructive interference between objectives, as well as enables us to formulate objectives over arbitrarily long time horizons, unlocking high quality estimates.

## 4 EULERFLOW

*Scene Flow via ODE* proposes a framework where the neural prior $\theta$ represents an estimate of the Eulerian flow field $F$ (i.e. $F \approx \theta$); however, it does not prescribe the optimization objectives for $\theta$. Thus, we instantiate Scene Flow via ODE with *EulerFlow*, a point cloud only scene flow method[2] with reconstruction and cyclic consistency objectives across the entire sequence of observations.

As we show in Equation 17 (Appendix C.4), we can use $\theta$'s Eulerian flow field estimate to extract an estimated point trajectory from $x_t, y_t, z_t$ at $t$ to some future location at time $t'$ via Euler integration over $\theta$ without requiring a canonical frame definition, i.e. $E_\theta(x_t, y_t, z_t, t, t')$. By extracting point trajectories for every point $p$ in $P_t$ using $E_\theta$, we can not only construct a two-frame scene flow estimate of $\mathcal{F}_{t,t+1}$, but also estimate flow to arbitrary future or prior timesteps (e.g. $\mathcal{F}_{t,t+2}$ or $\mathcal{F}_{t,t-1}$). This allows us to optimize over multi-frame reconstruction objectives: we can now pose reconstruction surrogate objectives between *any* two point clouds in our observation sequence, not just adjacent point clouds $P_t$ and $P_{t+1}$. Similarly, we can straightforwardly pose cyclic consistency objectives by composing $\mathcal{F}_{t,t+1}$ and $\mathcal{F}_{t+1,t}$. Formally, for $P_t$'s $\mathcal{F}_{t,t+k}$ (for any $k \in \mathbb{Z}$), we define

$$\text{Euler}_\theta\left(P_t, k\right) = P_t + \mathcal{F}_{t,t+k} = \forall p \in P_t : E_\theta(p_x, p_y, p_z, t, t+k) \ , \tag{2}$$

enabling us to pose $\theta$'s optimization objective $\forall P_t \in (P_0, \ldots, P_N)$ across the window of size $W$

$$\arg\min_\theta \sum \begin{array}{l} \forall k \in \{-W, \ldots, W\} \setminus \{0\} : \text{TruncatedChamfer}(\text{Euler}_\theta\left(P_t, k\right), P_{t+k}) \\ \alpha \left\|\text{Euler}_\theta\left(\text{Euler}_\theta\left(P_t, 1\right), -1\right) - P_t\right\|_2 \end{array} \tag{3}$$

In practice, we set W to 3 and $\alpha$ to 0.01. We provide additional implementation details in Appendix C. In order to optimize $\theta$, our estimate of the Eulerian flow field $F$, we perform Euler integration to extract point cloud flow estimates as part of reconstruction losses. Notably, EulerFlow only requires a single optimization loop over a single neural prior $\theta$ compared to NSFP's two priors $\theta$ and $\theta'$. Our neural prior $\theta$ is a straightforward extension to NSFP's coordinate network prior. Like with NSFP, TruncatedChamfer is defined as the standard $L_2$ Chamfer distance with per-point distances below 2 meters. As we show in Section 5, EulerFlow's simple ODE estimation formulation across multiple observations produces high quality flow, and solving this ODE over arbitrary time spans unlocks emergent point tracking behavior.

## 5 EXPERIMENTS

In order to validate EulerFlow's construction and better understand the impact of its design choices, we perform extensive experiments on the Argoverse 2 (Wilson et al., 2021) and Waymo Open (Sun et al., 2020) autonomous vehicle datasets. We compare against open source implementations of FastNSF (Li et al., 2023), Liu et al., NSFP (Li et al., 2021b), FastFlow3D (Jund et al., 2021), and variants of ZeroFlow (Vedder et al., 2024) provided by the ZeroFlow model zoo[3], a third-party implementation of NTP (Wang et al., 2022a) from Vidanapathirana et al., and Argoverse 2 2024 Scene Flow Challenge leaderboard submission results from the authors of Flow4D (Kim et al., 2024), TrackFlow (Khatri et al., 2024), DeFlow++/DeFlow (Zhang et al., 2024a), ICP Flow (Lin & Caesar, 2024), and SeFlow (Zhang et al., 2024b). As discussed in Khatri et al. and used in the Argoverse 2 2024 Scene Flow Challenge, methods are ranked by their speed normalized *mean Dynamic Normalized EPE*.

**Implementation Details.** To showcase the flexibility of EulerFlow without hyperparameter tuning, for all quantitative experiments we run with a neural prior of depth 8 (NSFP's default depth), except for our submission to the Argoverse 2 2024 Scene Flow Challenge (Section 5.1) where, based on our depth ablation study on the val split (Section 5.2.3), we set the depth of the neural prior to 18. As discussed in NTP's original paper (Wang et al., 2022a) and confirmed by our experiments, NTP struggles to converge beyond 25 frames, so we only compare against it in a 20 frame settings. As is

---

[2]Visualizations shown in color for better viewing. EulerFlow can also use monocular depth estimates (Appendix A.2)

[3]https://github.com/kylevedder/SceneFlowZoo, from Vedder et al. (2024).

typical in the scene flow literature (Chodosh et al., 2023), we perform ego compensation and ground point removal on both Argoverse 2 and Waymo Open using the dataset provided map and ego pose.

## 5.1 How does EulerFlow compare to prior art on real data?

EulerFlow achieves **state-of-the-art** performance on the *Argoverse 2 2024 Scene Flow Challenge* leaderboard. Despite being unsupervised, EulerFlow **surpasses *all* prior art, supervised or unsupervised**, including Flow4D (Kim et al., 2024)[4] and ICP Flow (Lin & Caesar, 2024)[5]. Notably, EulerFlow achieves $< 2.5\times$ lower error mean Dynamic EPE than ICP Flow and beats Flow4D by over 10%.

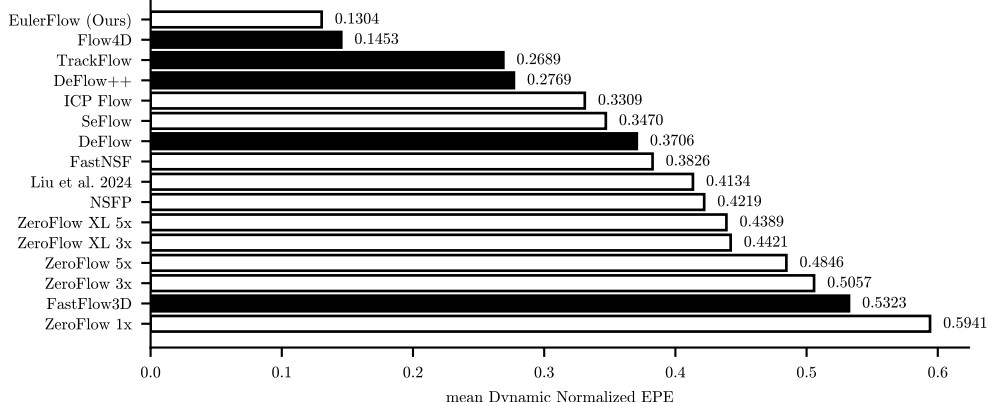

Figure 5: Mean Dynamic Normalized EPE of EulerFlow compared to prior art on the Argoverse 2 2024 Scene Flow Challenge test set. EulerFlow is state-of-the-art, beating all supervised (shown in black) and unsupervised (shown in white) methods. Lower is better.

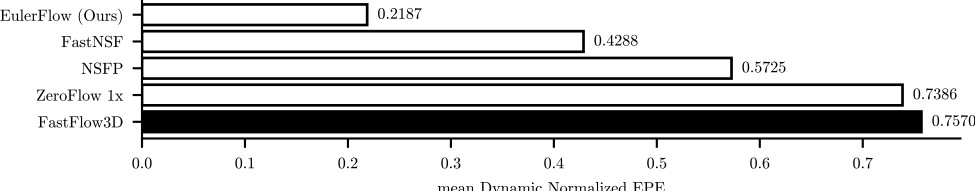

Figure 6: Mean Dynamic Normalized EPE of EulerFlow compared to prior art on the Waymo Open validation set. EulerFlow is state-of-the-art, beating all supervised (shown in black) and unsupervised (shown in white) methods. Lower is better.

EulerFlow's dominant performance also holds on Waymo Open (Sun et al., 2020); we compare against several popular methods (Figure 6), and EulerFlow again out-performs the baselines by a wide margin, more than halving the error over the next best method.

## 5.2 What contributes to EulerFlow's state-of-the-art performance?

We find that EulerFlow's lower mean Dynamic EPE can be attributed to better performance on smaller objects. On Argoverse 2, compared to Flow4D, EulerFlow's improves on `WHEELED VRU` (Figure 7d), a small, rare, fast moving class. Compared to ICP Flow, EulerFlow's improves on all classes (at least halving the error on every class!), with the largest improvements coming from the smaller and harder to detect objects `PEDESTRAIN` and `WHEELED VRU` (Figures 7c–7d). On Waymo Open, the same story holds; the most dramatic performance improvements come from the small object classes of `CYCLIST` and `PEDESTRIAN` (Figure 8).

These results are consistent with our qualitative visualizations. Figure 13 shows EulerFlow is able to cleanly extract the motion of a bird flying past the ego vehicle. Euler integration using EulerFlow's

---

[4]Flow4D is the winner of the 2024 Argoverse 2 Scene Flow Challenge supervised track.
[5]ICP Flow is the winner of the 2024 Argoverse 2 Scene Flow Challenge unsupervised track.

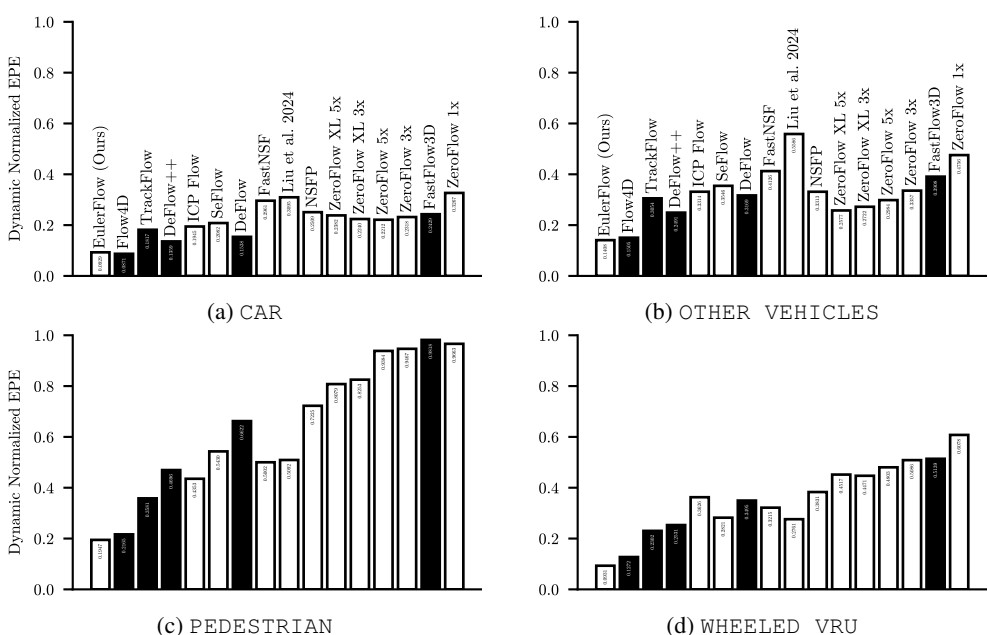

Figure 7: Per class Dynamic Normalized EPE of EulerFlow compared to prior art on the Argoverse 2 2024 Scene Flow Challenge test set. Supervised methods shown in black, unsupervised methods shown in white. Methods are ordered left to right by increasing mean Dynamic Normalized EPE. Lower is better.

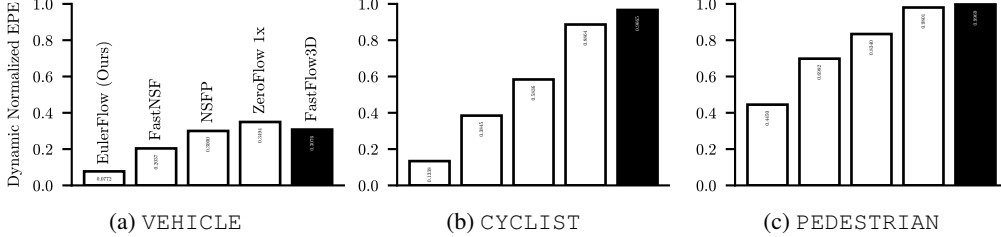

Figure 8: Per class Dynamic Normalized EPE of EulerFlow compared to prior art on the Waymo Open validation set. Supervised methods shown in black, unsupervised methods shown in white. Methods are ordered left to right by increasing mean Dynamic Normalized EPE. Lower is better.

ODE, starting at the bird's takeoff position and ending when it loses lidar returns, produces emergent 3D point tracking behavior on the bird through its trajectory (Figure 9), further demonstrating the quality of EulerFlow's model of the scene's motion.

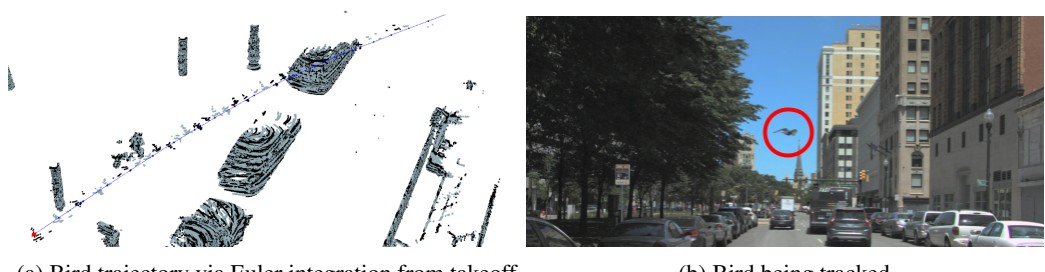

(a) Bird trajectory via Euler integration from takeoff          (b) Bird being tracked

Figure 9: EulerFlow is able to track the bird over 20 frames by performing Euler integration starting from takeoff until it loses all point cloud lidar returns.

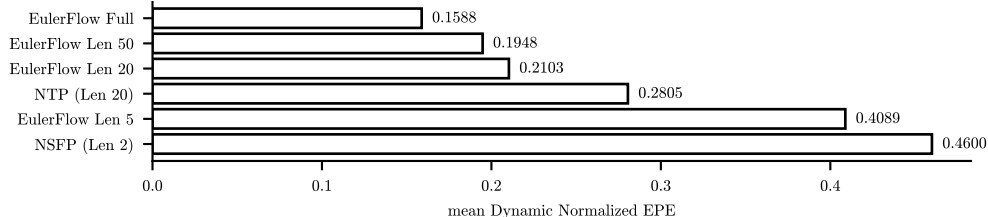

Figure 10: Mean Dynamic Normalized EPE of EulerFlow for various sequence lengths on the Argoverse 2 val split, compared against representative baselines. These results demonstrate that EulerFlow improves with sequence length; however, at a sequence length of 20, our method significantly outperforms NTP, suggesting that EulerFlow's performance cannot solely be attributed to longer sequence modeling.

### 5.2.1 HOW DOES OBSERVATION SEQUENCE LENGTH IMPACT EULERFLOW?

As we discuss in Section 3, EulerFlow benefits from constructive interference from ODE estimation over many observations. Does this sufficiently explain EulerFlow's performance? Figure 10 shows the performance of EulerFlow at length 5, 20, 50, and full sequence (roughly 160 frames) compared to NSFP and NTP at length 20. EulerFlow sees clear continual improvements as the number of frames increases without signs of saturation. However, sequence length alone does not explain EulerFlow's performance; even at the same sequence length of 20, EulerFlow demonstrates significantly better performance than NTP.

### 5.2.2 HOW DO MULTI-FRAME OPTIMIZATION OBJECTIVES IMPACT EULERFLOW?

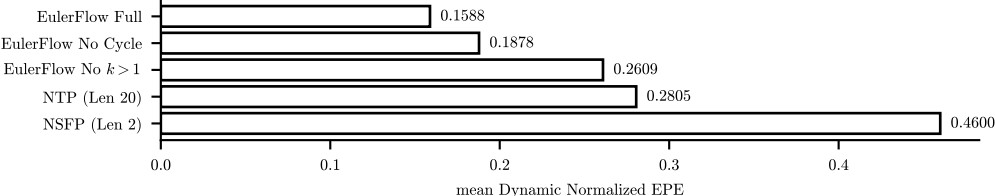

Figure 11: Mean Dynamic Normalized EPE of EulerFlow for various losses on the Argoverse 2 val split, compared against representative baselines. These results demonstrate that EulerFlow's multi-observation optimization objectives significantly improve overall performance.

Equation 3 outlines two major components of EulerFlow's loss: multi-frame Euler integration for Chamfer Distance reconstruction, and cycle consistency. Figure 11 compares EulerFlow without more than one integration step (No $k > 1$) and without cycle consistency rollouts (No Cycle) to better understand the impact of these components. Ablating multi-step Euler integrated rollouts results in significant degredation, as they are a strong forcing function to have consistent, smooth flow volumes; indeed, despite consuming the entire sequence, EulerFlow (No $k > 1$) is only slightly better than NTP with a sequence length of 20. These results highlight the power of multi-step rollouts and their potential as a objective for other test-time optimization methods and feedforward methods.

### 5.2.3 HOW DOES THE CAPACITY OF THE NEURAL PRIOR IMPACT EULERFLOW?

Li et al. ablate the capacity of NSFP's neural prior to characterize underfitting and overfitting to optimization objective noise, ultimately selecting a depth of 8. EulerFlow's neural prior is structured similarly; however, NSFP is fitting a single snapshot in time, while EulerFlow is fitting an entire ODE over significant time intervals. Intuitively, one would expect that full sequence modeling would benefit from greater network capacity.

To evaluate this, we perform a sweep of EulerFlow's network depth on the Argoverse 2 validation split (Figure 12). While EulerFlow with NSFP's default of depth 8 performs well on our Argoverse 2 evaluations (0.1% worse than the supervised state-of-the-art Flow4D), we see that performance improves as the neural prior's depth increases until depth 18 (indicating underfitting), where we start

to see degradation (indicating overfitting to noise). Based on these results our Argoverse 2 2024 Scene Flow Challenge leaderboard submission uses a depth 18 neural prior (Figure 5).

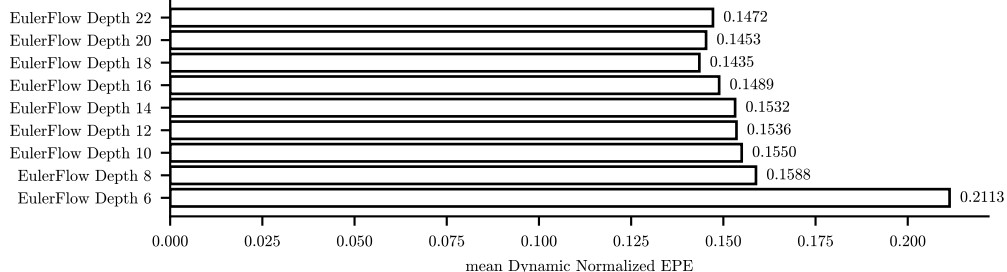

Figure 12: Mean Dynamic Normalized EPE of EulerFlow on the Argoverse 2 val split for different neural prior capacities. Shallow networks underfit the ODE, while deeper networks overfit to noise in the optimization objectives.

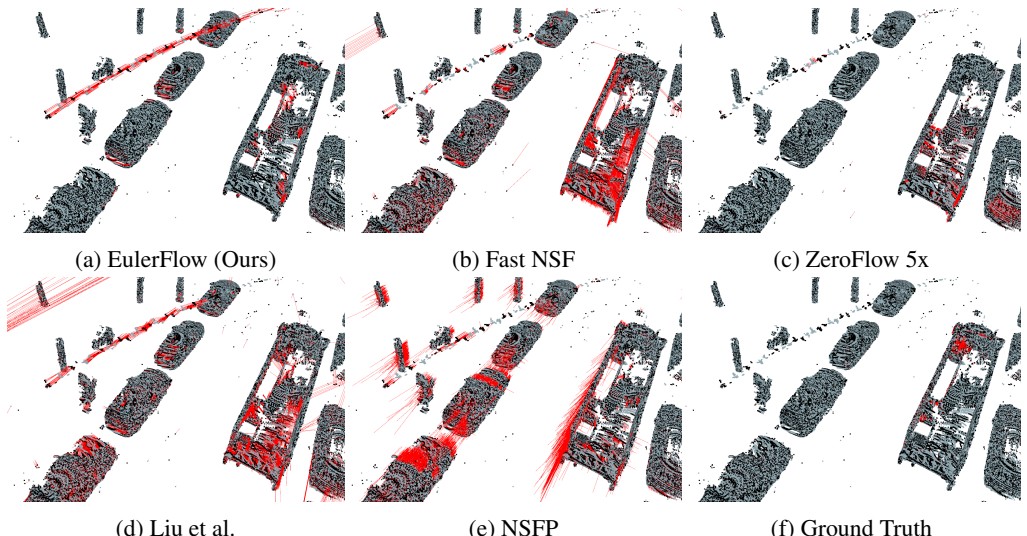

(a) EulerFlow (Ours)    (b) Fast NSF    (c) ZeroFlow 5x

(d) Liu et al.    (e) NSFP    (f) Ground Truth

Figure 13: Visualization of EulerFlow compared to prior art for the same scene as Figure 1a and Figure 9a. EulerFlow is able to extract the bird's trajectory; however, all other methods except Liu et al. fail to recognize this motion, and Liu et al.'s flow is marred by severe scene artifacts. The bird is outside the labeled object taxonomy, and so its motion is unlabeled in the ground truth (Figure 13f).

## 5.3 BEYOND AUTONOMOUS VEHICLES

Due to a dearth of real-world, labeled scene flow data, prior scene flow work on real data overwhelmingly evaluates on autonomous vehicle datasets (Dewan et al., 2016; Li et al., 2021b; Jund et al., 2021; Li et al., 2023; Chodosh et al., 2023; Liu et al., 2024; Vedder et al., 2024; Khatri et al., 2024); consequently, motion understanding in other important domains like tabletop manipulation has been neglected. To showcase EulerFlow's out-of-the-box flexibility and generalizability, we visualize EulerFlow on several dynamic tabletop scenes we collected using the ORBBEC Astra, a low cost depth camera commonly used in robotics (Figure 14). For viewing ease, we paint our point clouds with color; however, RGB information is not provided to EulerFlow during optimization. While EulerFlow only reasons about point clouds, it can leverage video mono depth estimates to describe RGB-only scene flow (Appendix A.2). Interactive visuals are available at `vedder.io/eulerflow`.

## 6 CONCLUSION

By reframing scene flow as fitting an ODE over positions for a full sequence of observations, we are able to construct EulerFlow, a simple unsupervised scene flow method that achieves state-of-the-art performance on the Argoverse 2 2024 Scene Flow Challenge and Waymo Scene Flow benchmark,

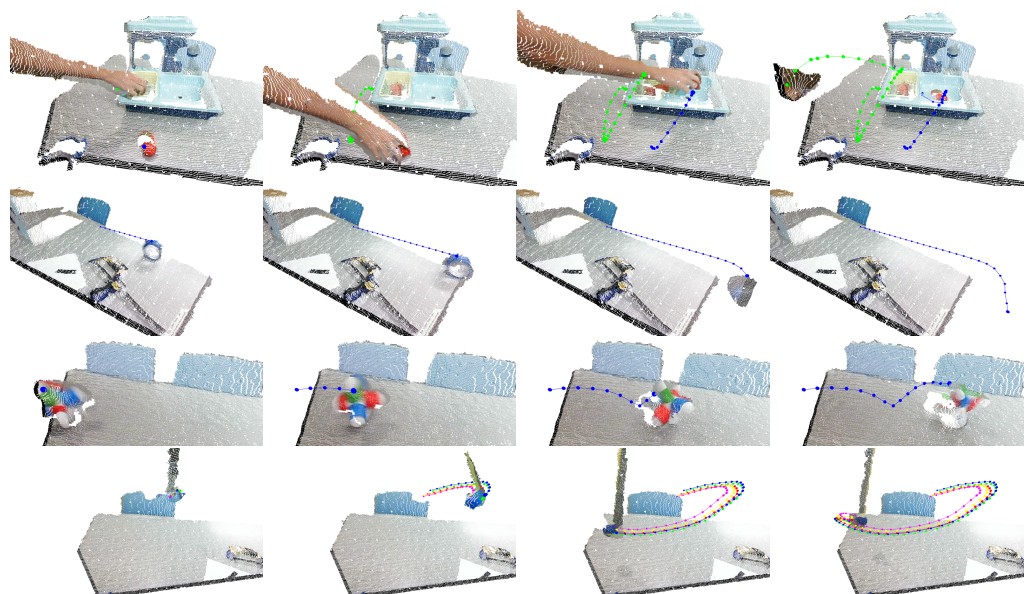

Figure 14: Visualizations of EulerFlow's emergent 3D point tracking behavior that demonstrate the quality of its ODE estimate. Row 1 depicts tracking a tomato placed in the sink by a human hand; note the point does not move despite the hand grasping the tomato. Row 2 depicts tracking of painters tape rolling off a table; EulerFlow is able to estimate its trajectory even after it disappears out of frame. Row 3 depicts tracking of the motion of a jack commonly used in tabletop manipulation experiments (Venkatesh et al., 2023). Row 4 depicts tracking of a tennis ball taped to a flexible rod. All tracks are produced by Euler integration through the estimated ODE from the initial conditions shown in the left column. Note that point clouds are shown in color for visualization purposes only.

where it beats all prior art, supervised or unsupervised. EulerFlow is able to describe motion on small, fast moving, out of distribution objects unable to be captured by prior art, suggesting that it makes good on the promises of scene flow as a powerful primitive for understanding the dynamic world. It also exhibits other emergent capabilities, like basic 3D point tracking behavior.

We believe that this ODE formulation has implications for scene flow at large, including beyond test-time optimization methods; the power of multi-step Euler integration may translate to feedforward network training. Future work should explore feedforward models that perform autoregressive rollouts or directly learn to estimate multiple steps into the future.

### 6.1 LIMITATIONS AND FUTURE WORK

EulerFlow's strong performance opens the book on an exciting new line of work; however, we feel that it's important to be candid about EulerFlow's current limitations in order to make future progress.

*EulerFlow is point cloud only.* Point cloud sparsity bottlenecks performance; for instance, in Figure 9 and Figure 13 we were only able to track the bird for 20 frames because we lost lidar observations of the bird, while it remained visible in the car's RGB cameras. Future works should explore multi-modal fusion for better long-term motion descriptions.

*EulerFlow is expensive to optimize.* With our implementation, optimizing EulerFlow for a single Argoverse 2 sequence takes 24 hours on one NVIDIA V100 16GB GPU, putting it on par with the original NeRF paper's computation expense (Mildenhall et al., 2021). However, like with NeRF, we believe algorithmic, optimization, and engineering improvements can significantly reduce runtime.

*EulerFlow does not understand ray casting geometry.* During ego-motion, a static foreground occluding object casts a moving shadow on the background; this causes Chamfer Distance to estimate this as a leading edge of moving structure, encouraging false motion artifacts (Li et al., 2021b). This can be addressed with optimization losses that model point clouds as originating from a time of flight sensor with limited visibility, as has been successfully demonstrated in the reconstruction (Chodosh et al., 2024) and forecasting literature (Khurana et al., 2023; Agro et al., 2024), rather than an unstructured set of points to be associated via local point distance.

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

# A   ADDITIONAL RESULTS

## A.1   HOW DOES THE CHOICE OF LEARNABLE FUNCTION CLASS AND DESIGN OF ENCODINGS IMPACT EULERFLOW?

EulerFlow at its core is an optimization loop over a simple, feedforward ReLU-based multi-layer perception inherited from Neural Scene Flow Prior (Li et al., 2021b). How does this choice of learnable function class impact the performance of EulerFlow? To better understand these design choices we examine the choice of non-linearity and time feature encoding.

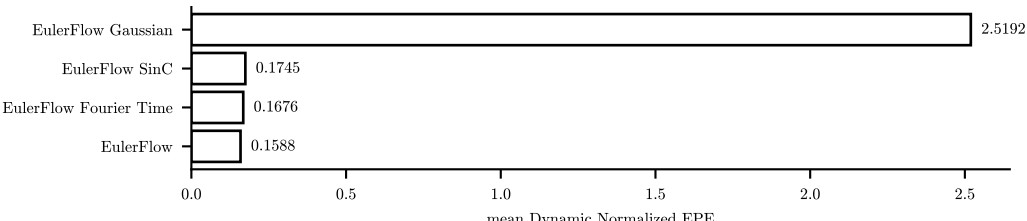

Figure 15: Mean Dynamic Normalized EPE of EulerFlow on the Argoverse 2 val split for less-smooth configurations of its learnable function class. These results indicate that the smoothness of the ReLU non-linearity proposed by Li et al. transfers well to EulerFlow.

One of Li et al.'s core theoretical contributions demonstrates that NSFP's ReLU MLP is a good prior for scene flow because it represents a smooth learnable function class, and scene flow is often locally smooth with respect to input position. However, unlike NSFP, EulerFlow is fitting flow over a full ODE; while it seems reasonable to assume that this ODE is typically also locally smooth, cases like adjacent cars moving rapidly in opposite directions may benefit from the ability to model higher frequency, less locally smooth functions. To test this hypothesis, we ablate EulerFlow by replacing its normalized time with higher frequency sinusoidal time embeddings (mirroring Wang et al.'s proposed time embedding for NTP), as well as try other popular non-linearities like SinC (Ramasinghe et al., 2024) and Gaussian (Chng et al., 2022) from the coordinate network literature. Figure 15 features negative results on these ablations across the board; Gaussians were unable to converge due the extremely high frequency representation triggering early stopping, while the use of SinC and higher frequency time embeddings both resulted in worse overall performance, indicating that Li et al.'s smooth function prior does indeed seem appropriate for EulerFlow's neural prior.

## A.2   EULERFLOW WITH MONOCULAR DEPTH ESTIMATES

While EulerFlow only consumes point clouds, we can leverage RGB-based video monocular depth estimators to fit scene flow. In Figure 16, we use DepthCrafter (Hu et al., 2024) to generate a point cloud from the raw RGB of the tabletop video from Figure 14, Row 4.

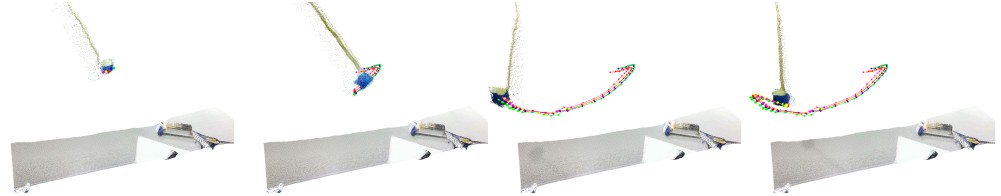

Figure 16: Visualizations of EulerFlow's emergent 3D point tracking behavior on monocular depth estimates from DepthCrafter (Hu et al., 2024). Interactive visualizations available at `vedder.io/eulerflow`.

## A.3   HOW DOES EULERFLOW FAIL?

As we discuss in Section 6.1, EulerFlow does not understand projective geometry — its optimization losses use Chamfer Distance which directly associates points, sometimes resulting in moving shadows

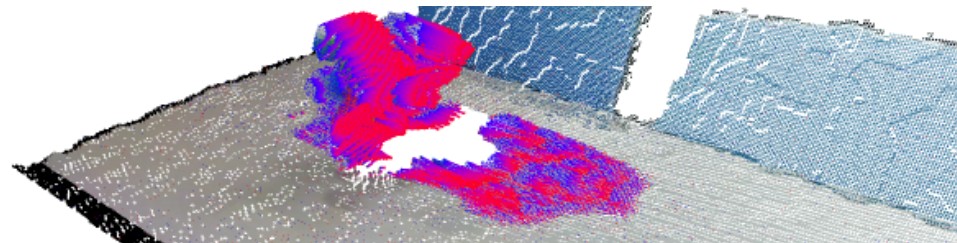

Figure 17: Visualizations of one of the failure modes of EulerFlow where flow is predicted on the edges of the moving "shadow" in the point cloud. Interactive visualizations available at `vedder.io/eulerflow`.

on background objects. To demonstrate this, we select a particularly egregious example in Figure 17, featuring a frame from the jack being thrown across the table. Due to the moving shadow cast by the jack onto the table, EulerFlow incorrectly assigns flow to the table surface nearby the jack, particularly on the leading edge, even though the table surface is stationary.

## B  FAQ

### B.1  WHAT DATASETS DID YOU PRETRAIN ON?

EulerFlow is not pretrained on any datasets. It is a test-time optimization method (akin to NeRFs), and as we show with our tabletop data, this means it runs out-of-the-box on arbitrary point cloud data.

### B.2  WHY DIDN'T YOU USE A NEURAL ODE OR A LIQUID NEURAL NETWORK?

Neural ODEs (Chen et al., 2018) take variable size and number of steps in latent space to do inference; imagine a ResNet that can use an ODE solver to dynamically scale the impact of the residual block, as well as decide the number of residual blocks. They are not a function class specially designed to fit derivative estimates well. Similar to Neural ODEs, Liquid Neural Networks (Hasani et al., 2021) focus on the same class of problems and are similarly not applicable.

### B.3  WHY DIDN'T YOU DO EXPERIMENTS ON FLYINGTHINGS3D / <SIMULATED DATASET>?

Most popular synthetic datasets do not contain long observation sequences (Mayer et al., 2016; Butler et al., 2012), but instead include standalone frame pairs. Our method leverages the long sequence of observations to refine our neural estimate of the true ODE. Indeed, on two frames, EulerFlow collapses to NSFP.

More importantly, these datasets are also not representative of real world environments. To quote Chodosh et al.: "[FlyingThings3D has] unrealistic rates of dynamic motion, unrealistic correspondences, and unrealistic sampling patterns. As a result, progress on these benchmarks is misleading and may cause researchers to focus on the wrong problems." Khatri et al. also make this point by highlighting the importance of meaningfully breaking down the object distribution during evaluation identify performance on rare safety-critical categories. FlyingThings3D does not have meaningful semantics; it's not obvious what things even matter or how to appropriately break down the scene.

Instead, we want to turn our attention to the sort of workloads that *do* clearly matter — describing motion in domains like manipulation or autonomous vehicles, where it seems clear that scene flow, if solved, will serve as powerful primitive for downstream systems. This is why we performed qualitative experiments on the tabletop data we collected ourselves; to our knowledge, no real-world dynamic datasets of this nature exist with ground truth annotations, but we want to emphasize that EulerFlow works in such domains, and consequently EulerFlow and other Scene Flow via ODE-based methods can be used as a primitive in these real world domains.

## C    EULERFLOW IMPLEMENTATION DETAILS

Our neural prior $\theta$ is a straightforward extension to NSFP's coordinate network prior[6]; however, instead of taking a 3D space vector (positions $X, Y, Z \in \mathbb{R}$) as input, we encode a 5D space-time-direction vector: positions $X, Y, Z, \in \mathbb{R}$, sequence normalized time $t \in [-1, 1]$ (i.e. the point cloud time scaled to this range), and direction $d \in \{\texttt{BWD} = -1, \texttt{FWD} = 1\}$. This simple encoding scheme enables description of arbitrary regions of the ODE, allowing for the ODE to be queried at frequencies different from the sensor frame rate. Euler integration enables simple implementation of multi-step forward, backward, and cyclic consistency losses without extra bells and whistles. For efficiency, we use Euler integration with $\Delta t$ set as the time between observations for our ODE solver, enabling support for arbitrary sensor frame rates, and set the cycle consistency balancing term $\alpha = 0.01$ and optimization window $W = 3$ for all experiments.

EulerFlow's definition of TruncatedChamfer is symmetric[7], i.e. TruncatedChamfer$(A, B)$ = TruncatedChamfer$(B, A)$. If this symmetric TruncatedChamfer is naively implemented via a performant differentiable CUDA accelerated K=1NN computation[8] from $A$ to $B$ and from $B$ to $A$, on an NVIDIA V100, EulerFlow spends roughly 80% of its compute time performing these KNN checks. To accelerate this, we precompute exact GPU accelerated KD-Trees (Grandits et al., 2021) for the input point clouds $\{P_0, \ldots, P_N\}$, and when possible query those trees instead of computing K=1NN. In practice, we found these queries are almost instant, and reduce the time spent computing K=1NNs to about 40% of the total wall-clock time of EulerFlow.

Additionally, while Equation 3 is phrased as independent Euler integration steps for each timestep, we are able to share integration across the losses; we perform two integrations from $t$ to $t + W$ and $t$ to $t - W$, and use the intermediary locations along this trajectory as inputs to intermediary losses.

### C.1    FORMULATING THE ODE

Given a (possibly moving) particle in some canonical frame (i.e. time 0), we define a function $L(x_0, y_0, z_0, t)$ that can describe its location at an arbitrary future time $t$, i.e. a Lagrangian description of motion (Figure 4).

$$L(x_0, y_0, z_0, t) = x_t, y_t, z_t \tag{4}$$

For notational clarity to access $x_t, y_t, z_t$ individually, we can define

$$L_x(x_0, y_0, z_0, t) = x_t \tag{5}$$
$$L_y(x_0, y_0, z_0, t) = y_t \tag{6}$$
$$L_z(x_0, y_0, z_0, t) = z_t \tag{7}$$

Similarly, we can define $F(x_t, y_t, z_t, t)$ to describe the instantaneous velocity of a point $x_t, y_t, z_t$ at some arbitrary time $t$, i.e. a Eulerian description of motion (Figure 4).

$$\frac{dL(x_0, y_0, z_0, t)}{dt} = \frac{dL}{dt} = \left( \frac{dL_x}{dt}, \frac{dL_y}{dt}, \frac{dL_z}{dt} \right) = F(x_t, y_t, z_t, t) \tag{8}$$

$F$ is defined in terms of the total derivative of $L$ with respect to $t$, as $x_0, y_0, z_0$ are initial conditions that do not vary with time (i.e. $\frac{dL}{dt} = \frac{\partial L}{\partial t} + \frac{\partial L}{\partial x_0}\frac{dx_0}{dt} + \frac{\partial L}{\partial y_0}\frac{dy_0}{dt} + \frac{\partial L}{\partial z_0}\frac{dz_0}{dt} = \frac{\partial L}{\partial t}$, as $\frac{dx_0}{dt} = \frac{dy_0}{dt} = \frac{dz_0}{dt} = 0$). We can exactly define $L$ recursively in terms of the initial conditions and $F$, i.e.

---

[6]Hyperparameters (e.g. filter width of 128) of NSFP's prior are kept fixed, except for depth (Section 5.2.3).

[7]This is in keeping with NSFP, but in opposition to other methods like FastNSF (Li et al., 2023). We found that a symmetric definition provided non-trivial performance improvements to EulerFlow.

[8]We used PyTorch3D (Ravi et al., 2020), which has custom CUDA operations with CUDA templated support for single neighbor differentiable KNN.

$$L(x_0, y_0, z_0, t) = (x_0, y_0, z_0) + \int_0^t F(L_x(x_0, y_0, z_0, \tau), L_y(x_0, y_0, z_0, \tau), L_z(x_0, y_0, z_0, \tau), \tau) d\tau \tag{9}$$

or, more compactly,

$$L(x_0, y_0, z_0, t) = (x_0, y_0, z_0) + \int_0^t F(x_\tau, y_\tau, z_\tau, \tau) d\tau \tag{10}$$

Our function $L$ can thus be defined as a multi-dimensional ODE in terms of $F$ with initial conditions $x_0, y_0, z_0$.

## C.2 ARBITRARY START AND END TIMES FROM THE EULERIAN FORMULATION

In the above derivation, $L$ requires that a moving point be defined in terms of a canonical frame defined at time 0, as is common in the deformation in reconstruction literature. However, the Eulerian formulation has no such requirement, allowing us to select arbitrary start and end times across different point queries. To showcase this, we can query $F$ to extract the trajectory of a particle at $t$ across the range $[t, t']$ starting at $x_t, y_t, z_t$ simply by changing the range of the integral in Equation 10, i.e.

$$E(x_t, y_t, z_t, t, t') = (x_t, y_t, z_t) + \int_t^{t'} F(x_\tau, y_\tau, z_\tau, \tau) d\tau \tag{11}$$

While $E$ and $L$ appear similar on their face, $E$ is strictly more flexible than $L$. In principle you could choose to redefine $L$ to use $t$ as the time for your canonical frame, but this is a *global* choice; you cannot do this on a per-query basis. However, with $E$'s Eulerian framing, we can extract a different point's trajectory from the entirely different range $t^\dagger$ to $t^\ddagger$ (i.e. $E(x_{t^\dagger}, y_{t^\dagger}, z_{t^\dagger}, t^\dagger, t^\ddagger)$) without concern for a canonical frame definition. It need not even be the case that $t < t'$; indeed, this extraction works even if $t > t'$, i.e. extracting the backwards trajectory through time.

## C.3 EULER INTEGRATION TO APPROXIMATELY SOLVE THE ODE

If $F$ is of arbitrary form and we want to compute the concrete values of $L$, we cannot exactly compute the continuous integral from 0 to $t$; we must approximate this with finite differences. Thus, we split the time range 0 to $t$ into $k$ steps, where each step is of size $\frac{t}{k}$. Thus, we can again define $L$ via recursion, but this time explicitly.

$$L(x_0, y_0, z_0, 0) = (x_0, y_0, z_0) \tag{12}$$

$$L(x_0, y_0, z_0, \tau + \frac{t}{k}) \approx L(x_0, y_0, z_0, \tau) + \frac{t}{k} \cdot F(x_\tau, y_\tau, z_\tau, \tau), \tag{13}$$

or directly without recursion,

$$L(x_0, y_0, z_0, t) \approx (x_0, y_0, z_0) + \sum_{n=1}^{k} \frac{t}{k} \cdot F(x_{n\frac{t}{k}}, y_{n\frac{t}{k}}, z_{n\frac{t}{k}}, n\frac{t}{k}) \tag{14}$$

This finite difference solving approach is Euler integration.

## C.4 ESTIMATING THE FLOW FIELD WITH EULERFLOW'S NEURAL PRIOR

For a given scene, we do not have access to $L$ or $F$ directly; these are are the *true* functions that uniquely characterize the underlying motion of the scene that we are trying to estimate. For EulerFlow, we represent our estimate of the scene's flow field $F$ with a neural prior, $\theta$, i.e.

$$F(x, y, z, t) \approx \theta(x, y, z, t) \tag{15}$$

and thus

$$L(x_0, y_0, z_0, t) \approx (x_0, y_0, z_0) + \sum_{n=1}^{k} \frac{t}{k} \cdot \theta(x_{n\frac{t}{k}}, y_{n\frac{t}{k}}, z_{n\frac{t}{k}}, n\frac{t}{k}) \tag{16}$$

and, using the arbitrary start and end definition from Appendix C.2, with $k$ steps from the range $t$ to $t'$ and $\delta = \frac{t'-t}{k}$

$$E(x_t, y_t, z_t, t, t') \approx E_\theta(x_t, y_t, z_t, t, t') = (x_t, y_t, z_t) + \sum_{n=1}^{k} \delta \cdot \theta(x_{n\delta+t}, y_{n\delta+t}, z_{n\delta+t}, n\delta+t) \tag{17}$$

This formulation makes EulerFlow highly flexible, enabling optimization of $\theta$'s estimate of $F$ with objectives that take either an Eulerian view (directly on $\theta$ via Equation 15) or a Lagrangian view (on point rollouts for arbitrary start and end ranges via Equation 17).

