# OpenReview forum: "Neural Eulerian Scene Flow Fields"
_ICLR.cc/2025/Conference — ICLR 2025 Poster_

### Official Review · Reviewer_XQYd · 2024-10-25

**Soundness:** 3
**Presentation:** 2
**Contribution:** 3
**Rating:** 6
**Confidence:** 4

**Summary:**

The paper proposes SFvPDE, a framework to cast scene flow estimation as a PDE with a neural prior, and EulerFlow, an example demonstrating how SFvPDE can be trained using the Euler method to locally integrate the PDE during training. A space-time-dependent vector-field is trained to match subsequent point clouds at different timestamps via solving the underlying PDE. The method significantly outperformes both supervised and unsupervised baselines and is especially effective on small objects compared to prior work.

**Strengths:**

The method is simple and intuitive yet effective. Extensive experiments section shows clearly that the proposed method surpasses prior work.

**Weaknesses:**

The main weakness of EulerFlow, as also noted by the authors (lines 524-528), is the time it takes to converge on a single scene. But given the performance of the method, this should not be considered critical. However, the presentation of the paper can be improved, the paper lacks some implementation details, as from time to time the reader has to guess what is actually happening (see questions section).

**Questions:**

Here are some questions and concerns regarding the presentation and the method:

1) In lines 189 and 195 $\frac{\partial L^*}{\partial t}$ is referred to as the partial differential equation, or a PDE. However, $\frac{\partial L^*}{\partial t}$ alone is not a PDE yet, unless it is set equal to something (as in equation 2).
2) I guess that in equation 2 SFvPDE should also depend on $x$. Could the authors clarify this?
3) In general, it would be nice to have more formal definitions. E.g. in EulerFlow, an exact formula for solving the PDE, $\text{Euler}_\theta(P_t, d, k)$, would improve understanding and reproducibility of the method.
4) In principle, the PDE can be integrated in both directions by simply reverting the time. The usage of the direction as an extra argument in the model makes the connection between sections 3 and 4 slightly weaker and seems to be a legacy design choice from NSFP. Thus, a question to the authors is whether they have tried training without the direction argument?
5) Given the high computational complexity of the method, it would be better to see some implementation details on how exactly equation 3 is calculated during training. Are any optimizations already incorporated? E.g. in the current form separate terms in the loss are independent. However, I believe that subsequent Euler steps can use previous estimates instead of recalculating them.
6) More ablation studies would better highlight the contributions of the paper. E.g. how general and how sensitive is the method to different numerical solvers and sizes of discretization steps? Have the authors tried higher-order PDE solvers or using $\Delta t$ smaller than the time between observations?

I will adjust my score based on the other reviews and the rebuttal by the authors.

---

> ### Author Response · Authors · 2024-11-18
>
> Thanks for your review. We think your questions have surfaced several valuable points, and we have clarified and improved our presentation and discussion accordingly.
>
> **Sharpening the differential equation formalization**
>
> To sharpen our formalization, we have added a full derivation of the differential equation in Supplemental D. In D.1 we perform a derivation from the Laplacian definition of the field ($L$) to our Eulerian formulation of the field ($F$), ultimately presenting $L$ in terms of a differential equation using $F$. In D.2 we then formalize the ability to employ arbitrary start and end times with the diff eq, in D.3 we derive Euler integration over this diff eq to approximate it, and in D.4 we describe replacing $F$ by our neural approximation ($\Theta$). Using this derivation, we present a formal definition for $Euler_\Theta$ in the main text (Section 3, Equation 2).
>
> As part of a broader cleanup, we have updated the title and sharpened Section 3 and 4 to focus more on the theoretical construction and benefits of scene flow as a differential equation.
>
> **Implementation details**
>
> We have made EulerFlow’s implementation details its own section in Supplemental C in order to separate them from the presentation in Section 4. We have added detailed commentary on the implementation details of our various primitives, e.g. KDTree precomputation for some of the ChamferDistance KNN calls, including citations for the used libraries and commentary on relative runtime impacts, and sharing of Euler integration rollouts between steps in the main loss (Equation 3). As part of a push for reproducibility, we will release the code for EulerFlow upon publication.
>
> You are correct that the input of direction into $\Theta$ is a legacy design decision. We removed this discussion in the main text, as we do not believe it provides meaningful value to the method, but we describe it in Appendix C as a matter of transparency.
>
> **Different diff eq solver configurations**
>
> We think this is a great direction for future work. Unfortunately, a smaller step size during optimization proportionally increases computation — $\Delta t / 2$ requires twice as many steps as $\Delta t$ to express the same trajectory, resulting in greater runtime and greater VRAM usage for gradients.
>
> However, despite being optimized for $\Delta t$ steps, we can after-the-fact query an optimized EulerFlow representation using any arbitrary solver, including $\Delta t / 2, \Delta t / 4, \Delta t / 8$ Euler integration. We have updated our project page (http://eulerflow.github.io/) to visualize these solver trajectories on our interactive scenes. Qualitatively, these trajectories can sometimes be a bit better, but are just as often egregiously bad; this makes sense given the representation was not optimized to perform well with Euler integration under these settings.

---

> > ### Comment · Reviewer_XQYd · 2024-11-21
> > **Response to the Authors**
> >
> > Dear Authors,
> >
> > Thank you for your time and valuable additional clarifications in the rebuttal. I will keep my original positive rating.
> >
> > Best regards,
> > Reviewer

---

### Official Review · Reviewer_DftH · 2024-11-04

**Soundness:** 3
**Presentation:** 3
**Contribution:** 2
**Rating:** 6
**Confidence:** 4

**Summary:**

The paper proposes a neural representation to optimize scene flow as a discrete partial differential equation of scene geometry over time. Compared to previous method, Neural Scene Flow Prior (NSFP), a method is most related to this work in the neural representation, the proposed method introduces a multi-frame formulation and learns bi-directional three-step Euler integration of the geometry consistency using decoded per-frame scene flow. Compared to previous work, the proposed representation can achieve better performance in Autonomous driving datasets and the authors demonstrate qualitative performance on depth camera input as well.

**Strengths:**

1. The proposed scene flow representation is simple and technical sound. Compared to prior work NSFP, extending it to multi-frame and learns a bi-directional consistency scene flow is a very intuitive step forward.
2. The performance of this method (both qualitative and quantitative) is impressive. The method can learn very consistent scene flow in trajectory despite not explicitly considering common issues such as occlusion artifacts. As the paper demonstrated, it can tackle well on small objects (with potentially large motions as well).

**Weaknesses:**

1. The paper title and introduction is very general and does not provide a precise position of this paper's main contribution. "Scene flow a partial differential equation" has been historically formulated long time ago in many prior paper, e.g. [1] as one examplar reference, and it has been proposed as a continuous representation in one early seminal work [2]. Many related work studied this optimization problem using images input and solved it using differential optimization approaches before. In this paper seems only consider related work in the point cloud space, and beneficially solved in using a neural representation. I will suggests to more precise position their scope and contributions  in paper title, introduction and contributions.

2. The evaluation dataset in this paper is mostly on autonomous driving datasets though as the method demonstrated, it should also work on other data domain when depth is available. Though real world depth and flow ground  truth is hard to get, it won't be too hard if evaluated using a more general synthetic dataset that provide different motion patterns, compared to the type of motion and accuracy that autonomous driving dataset can provide.

3. The paper has already discussed the main limitations it section 6.1. Particularly for the last point "EulerFlow does not understand ray casting geometry", it was clear how this has been demonstrated in the current results. It will be good if the authors can provide examples and metrics that reflect the challenge in this aspect.

[1] A Variational Method for Scene Flow Estimation from Stereo Sequences, Frederic Huguet and Frederic Devernay, ICCV 2007

[2] Three dimensional scene flow, Vedula et al, CVPR 1999

**Questions:**

Among all the three points I illustrated in the weakness,

For point 1, I hope the authors can provide concise update on their paper title and contributions in particular for the first bullet time (line 99-100).

For point 2, the current evaluation is sound and maybe sufficient for this paper. I do believe it is nice to more evaluation on non-AV dataset quantitatively that very likely will benefit this method as a baseline for future work in different domain.

For point 3, it will be good if the author can provide specific example (as a figure)

---

> ### Author Response · Authors · 2024-11-18
>
> Thank you for your review. We have updated our draft to incorporate your feedback.
>
> **Title and framing are too general and insufficiently informative**
>
> We think this is very valuable feedback. Consequently, we have changed the title to _Neural Eulerian Scene Flow Fields_. We feel this better reflects the core of our contribution: of using neural representations to model  Eulerian flow fields (see Figure 4 in the updated draft). Unfortunately, while we have updated the PDF’s title, it appears we cannot update the OpenReview title during discussion, but we will be sure to change that for the camera ready.
>
> We have also sharpened our language around our contributions to make it clear that our novelty comes from formulating scene flow as a differential equation over _all observations_, which differentiates it from prior art including the cited seminal papers.
>
> **Evaluation on other (synthetic) datasets**
>
> To our knowledge, there aren’t good _real-world_ datasets for scene flow outside of Autonomous Vehicles. As we discuss in Supplemental B.3, in addition to often not having long observation sequences, Chodosh et al. [3] point out that synthetic data often has unrealistic scan patterns and object motion when compared to the real world.
>
> Consequently, while we are unable to provide quantitative results on the tabletop scene, we feel that the qualitative results (images in the paper and interactive demos on our project page) demonstrates EulerFlow’s value to domains beyond autonomous vehicles. That said, we, like you, strongly believe that this lack of quantification on real-world data outside of AV is a big limitation for the subfield, and this is an important area of future work to move the subfield forward – this subfield means relatively little if not merged into the broader field’s efforts towards building large intelligent systems.
>
> **Clearer depictions of method’s failures**
>
> We think this is great feedback. We have added Supplemental A.3 that focuses on further illustrating the failures we discuss in Section 6.1.
>
>
> [1] Rigid Scene Flow for 3D LiDAR Scans. Dewan et al. IROS 2016.
>
> [2] LaserNet: An Efficient Probabilistic 3D Object Detector for Autonomous Driving. Meyer et al. CVPR 2019.
>
> [3] Re-Evaluating LiDAR Scene Flow. Chodosh et al. WACV 2024.

---

> ### Comment · Reviewer_DftH · 2024-11-27
>
> Thanks for addressing my questions and concerns in weakness description and question sections. I don't have additional questions or concerns. The paper is very clear about its own limitations and will pave way for future work in this direction. The technical novelties and insights brought by this paper clearly push the state-of-the-art forward. I believe the work can have bigger impact if it can provide quantitative evaluation in additional scenarios and also agree this does not hurt the evaluation in this paper and can be left for future work.
>
> Given the position of the paper, I will remain the original rating above acceptance.

---

### Official Review · Reviewer_pt2w · 2024-11-04

**Soundness:** 3
**Presentation:** 3
**Contribution:** 3
**Rating:** 8
**Confidence:** 3

**Summary:**

The paper introduces a novel approach to scene flow estimation by reframing it as the task of estimating a continuous space-time partial differential equation (PDE) that describes motion across an entire observation sequence. The proposed method, called EulerFlow, utilizes a neural prior to represent this PDE and optimizes it against multi-observation reconstruction objectives. This approach allows for high-quality, unsupervised scene flow estimation on real-world data, outperforming existing supervised and unsupervised methods, particularly on small and fast-moving objects like birds and tennis balls. The authors demonstrate the effectiveness of EulerFlow on datasets such as Argoverse 2 and Waymo Open Dataset, and show its applicability across different domains without domain-specific tuning. Additionally, they highlight emergent 3D point tracking behavior as a result of solving the estimated PDE over long time horizons.

**Strengths:**

- I'm not up-to-date to the latest scene flow models, but from the results in the paper it surpass the prior art by a large margin, which is very significant
- Introducing the concept of modeling scene flow as a PDE is innovative and offers a new direction for research in motion estimation.
- The method is rigorously developed, with comprehensive experiments and ablation studies that validate the approach.
- The paper is well-written, with clear explanations and effective use of figures to illustrate key points.

**Weaknesses:**

- As stated in the paper, the speed of the proposed method is a big concern, preventing it from deploying on real world application.
- Some hyperparameters, such as the depth of the neural prior, seem to require dataset-specific tuning (e.g., increasing depth to 18 for the Argoverse 2 challenge), which may affect the method's out-of-the-box applicability.
- It would be great if the author could show more failure cases to help readers better understand its limitations.

**Questions:**

Overall I believe this paper is in a good shape, the authors discuss the properties and limitations of the proposed method thoroughly in the paper. I have a few more questions:

- How does the method handle scenes with deformable objects?
- What is the impact of temporal sampling rate on performance?
- How does the point cloud density affect the performance?

---

> ### Author Response · Authors · 2024-11-18
>
> Thank you for your review. We have updated our draft to incorporate your feedback.
>
> **Runtime of Eulerflow**
>
> We agree that, as currently implemented, the optimization speed is prohibitively slow and needs to be significantly improved to be deployed at any reasonable scale. As we point out in Section 6.1, the first NeRF method also took roughly 24 hours to optimize on a single scene, but follow up work on algorithmic, optimization, and engineering improvements have significantly reduced NeRF optimization time.
>
> To further substantiate this for EulerFlow, here's some back of the napkin math:
>
> EulerFlow is fully GPU compute constrained (i.e. not memory bandwidth constrained like some applications such as LLM training [1]) during optimization using V100s; given improvements in Float32 throughput, we can reasonably expect at least a roughly 3x speedup by switching to the latest GPUs [2], and with lower precision / sparsity exploration (V100s are Turing based, which have none of the Ampere / Hopper hardware acceleration features), etc we can feasibly get another 3x - 4x improvement.
>
> If, via a combination of above steps, we cut the runtime 10x, i.e. roughly 24 to 2.4 hours, we think we are in the ballpark of being feasible to scale up. ZeroFlow [3] performs distillation from NSFP into a feedforward student at scale; their cited numbers for NSFP are roughly 26 seconds per frame pair, roughly 1.11 hours per ~155 frame sequence. This is only about 2x faster than our hypothetical improved EulerFlow. At current open market cloud H100 prices (roughly 2.50 USD / card / hour), this would be roughly 6 USD / sequence. Given EulerFlow’s scene flow quality (and its ability to extract long tail flow on out-of-taxonomy objects like birds!), we feel this is very much worth the cost to get good quality pseudolabels.
>
> **Hyperparameters for EulerFlow**
>
> An attractive aspect of EulerFlow is, outside of cripplingly bad hyperparameter settings (Figure 11), it works out-of-the-box on new domains.
>
> For our tabletop demos, we took the exact same config for our Argoverse 2 experiments and just fed in our tabletop data and it worked on the first try. EulerFlow will almost certainly work better with domain-specific hyperparameters, but we find that reasonable settings (e.g. Depth 8 or 18) on a new domain work well.
>
> **Visual failure cases for EulerFlow**
>
> We think this is a great point; in Supplemental A.3 we have added a figure showcasing EulerFlow’s failure on the tabletop jack. In keeping with our discussion on EulerFlow's limitations (Section 6.1), we explain why this general class of failures is caused by the lack of ray casting geometry.
>
> **EulerFlow handling deformable objects?**
>
> We think this is a great question, as it highlights an important area of contribution: our method itself makes no assumptions about rigidity in the representation or loss functions, so in principle it’s able to handle deformable objects.
>
> We also demonstrate this in practice with movement of the hand in the _Hand Place in Sink_ scene and flexing of the rod in the _Tennis Ball on a Flexible Rod_ scene on our demo website (https://eulerflow.github.io/) — in both cases, the method is able to describe these motions without issue.
>
> **Point cloud density (space and time axes)**
>
> Point cloud density improves performance on both the space and time axes, as both enable Chamfer Distance to better approximate the true flow. NSFP discusses the failure cases of Chamfer Distance in the spatial axes; DynamicFusion [4] discusses the importance of having high frame rate sampling in making these reconstruction problems easier because motions are smaller and thus the interpolation problem is less challenging.
>
> [1] FlashAttention: Fast and Memory-Efficient Exact Attention with IO-Awareness. Dao et al. NeurIPS 2022.
>
> [2] Hopper Whitepaper, Nvidia, GTC 2022.
>
> [3] ZeroFlow: Scalable Scene Flow via Distillation. Vedder et al. ICLR 2024.
>
> [4] DynamicFusion: Reconstruction and Tracking of Non-rigid Scenes in Real-Time. Newcombe et al. CVPR 2015.

---

### Official Review · Reviewer_V3mD · 2024-11-05

**Soundness:** 3
**Presentation:** 2
**Contribution:** 3
**Rating:** 6
**Confidence:** 4

**Summary:**

This paper proposes to represent scene flow as a velocity field using a neural prior. Instead of prior art that directly represents per-pair scene flow as neural prior, the authors alternatively propose to use neural prior to model the partial differential equation (PDE) of the position of the point versus the time interval. This novel velocity representation is interesting and could offer flexibility in dealing with long-term sequences of flow estimations as the authors described in the paper. The authors have also done extensive analysis of the proposed method on Argoverse 2 (and Waymo) datasets, comparing the performance with recent scene flow works, and validating the good performance of the proposed method.

**Strengths:**

- The paper proposes an interesting idea to represent the scene flow as a velocity field using a neural network, making it very easy to combine the temporal information (time) and the spatial information (position of points).
- The authors have done extensive analysis of the proposed method, and have shown different ablation studies to validate the effectiveness of the method.
- The proposed method also shows the potential to deal with small objects and emergent flows in robotics scenarios, which could be interesting when applied to highly dynamic environments.
- Overall, the writing of the paper is clear, and the visualization is easy to understand.

**Weaknesses:**

- When using the time interval between [-1, 1] for the time encoding, will the proposed method not be able to handle time step outside the range? Given that the representation is a continuous neural network, how does it extrapolate to a longer sequence with the current representation?
- When comparing with a method like NSFP, I wondered if the authors could show the results of pure Euler integration of the method and highlight the benefits of wrapping a PDE with a neural network.
- The authors mentioned that they only do sequences of length 20, I wondered if the method failed rapidly with the increase of the sequence length. It would be interesting to show an even longer sequence to highlight the arbitrary time query property of the proposed method.
- I feel like the authors want to talk about too many things in this paper, so they may overlook the most important part of the method. This method is good at dealing with long-term flow trajectory and has the potential to better capture the small, highly dynamic objects in the scene. The authors could reorganize the motivations and experiments to highlight the advantages of the proposed method.

**Questions:**

- The discussion of the different activation functions (appendix) is indeed interesting. And this could be one of the interesting parts of the ablation study. However, it is strange to see that using the Gaussian non-linear function is yielding very bad performance. Perhaps the spectral width needs to be fine-tuned, especially when the distribution of the lidar scene flow is very unique.

Please also see the above section for detailed comments.

---

> ### Author Response · Authors · 2024-11-18
>
> Thank you for your review. We have updated our draft to incorporate your feedback.
>
> **Sequence length of only 20 / what happens if the sequence is longer?**
>
> By default, EulerFlow is optimized over the _full sequence_ (e.g. ~150 to 160 frames on Argoverse 2).
>
> The 20 frames setup you are referencing comes from a particular comparison of EulerFlow to NTP [1]: in order to perform a fair comparison against NTP, which is only designed to (and only performs well on) shorter trajectories, we evaluated EulerFlow and NTP head to head on these 20 frame subsequences. Figure 10 depicts these results, as well as the extension of EulerFlow to the full sequence length (where full length EulerFlow performs significantly better).
>
> **Time interval of [-1, 1]**
>
> EulerFlow is doing full sequence motion _reconstruction_, where -1 represents time at the beginning of the sequence and 1 represents time at the end of the sequence. Importantly, our method is not designed to do forecasting outside of the time range, in the same way a Dynamic NeRF is not designed to extrapolate beyond the time range of the given data.
>
> However, in principle, you could choose to query our differential equation beyond the optimized time horizon (e.g. at time 1.1 or -1.1) and you will receive some extrapolated estimate. To give some additional hint at what might happen when extrapolating, take a look at the point tracking on the tape from the _Ball and Tape_ scene on our demo website (the third scene from the left; https://eulerflow.github.io/)  — the point being tracked (the tape) disappears from the scene, resulting in point tracking across a region without data support; the point estimate just slowly drifts through space.
>
> **Euler integration between successive NSFP estimates**
>
> This baseline is in NTP's evaluations [1]. Unsurprisingly, this baseline is significantly worse than NTP (which EulerFlow in turn significantly outperforms), as NSFP is not optimized to provide multi-frame motion estimates,  so chaining outputs across frame pairs together produces catastrophic trajectory artifacts.
>
> **Organization improvements**
>
> Thank you for the suggestion. We have improved the organization Section 3 and 4 and have added additional implementation details to Appendix C.
>
> **Choice of non-linearity**
>
> We believe this is indeed a very interesting (and possibly very fruitful) line of future work. At the very least, we think it’s not on-its-face obvious that ReLUs are the “right” nonlinearity for scene flow, and a more careful and theoretically grounded study might find a better one that results in higher quality flow estimates. We chose to include these preliminary results in the Supplementary because they were an early experiment we ran that we thought might provide some signal to the community for what to look at next.
>
> In that vein, while we believe smoothness is an important factor for the ReLU MLP's good performance, we agree that the poor performance of the Gaussian is probably a consequence of poor hyperparameter tuning for spectral width. We believe the right prior plus some other tricks (e.g. learning an offset for the spectral width per layer) will allow it to converge, but we abandoned this effort for this project due to the good performance of the ReLU MLP.
>
> [1] Neural Prior for Trajectory Estimation. Wang et al, 2022

---

> > ### Comment · Reviewer_V3mD · 2024-11-26
> > **Response to authors' rebuttal**
> >
> > Thanks for providing a detailed response.
> >
> > I have read all the reviews and comments, I think most of the concerns were addressed by the authors. Therefore, I would like to keep my original positive score.

---

### Author Response · Authors · 2024-11-18

# Summary of Reviews

We present EulerFlow, an “interesting” (V3mD), “simple” (DftH), “innovative” (pt2w), and “effective” (XQYd) reframing of scene flow as the task of estimating a continuous space-time differential equation that describes motion across an entire observation sequence. We perform extensive analysis on Argoverse 2 and Waymo Open (V3mD, pt2w), outperforming both supervised and unsupervised prior art (V3mD). Notably, EulerFlow is particularly effective at estimating flow for small objects (pt2w, XQYd), and presents emergent 3D point tracking behavior (pt2w).

We want to thank the reviewers for their questions and comments, as we believe they have materially improved our paper presentation, including:

 - [DftH] updating the title of our paper to _Neural Eulerian Scene Flow Fields_ to better highlight our contribution to represent scene flow as a velocity field using a neural prior.

 - [XQYd, V3mD] updating Section 3 and 4 to improve presentation clarity / reproducibility and include additional implementation details in Appendix C and D. We will release our code to facilitate future work.

 - [pt2w] visualizing failure cases in Appendix A.3 to better illustrate our method’s limitations.

---

### Meta-Review · Area_Chair_eA7c · 2024-12-19

**Metareview:**

The paper presents EulerFlow, a framework for scene flow estimation over 3D point clouds, where EulerFlow is a neural network that produces the flow of a given point between two given time steps, the parameters of the network are optimized by minimizing the forward and cycle consistency errors. Experiments are presented on the ArgoVerse scene flow challenge, obtaining state-of-the-art results.

The paper received overall favorable reviews with one accept and three borderline accepts. The key strength of the paper is the large empirical improvements showcased on the ArgoVerse dataset, which was appreciated in many of the reviews.

**Additional Comments On Reviewer Discussion:**

The reviewers raised concerns on three major fronts:
* on the lack of clarity in the technical exposition (XQYd, V3mD),
* enormous compute needed to solve the scene flow problem per scene (pt2w, XQYd), and
* lack of experiments and ablation studies (DftH)

In the discussion that ensued, authors revised the paper, by changing the title to emphasize the main contribution, added technicalities explaining the formulation better, and fixed some mistakes. The concern of speed is an important one and cannot be rectified within the current setup.

Overall, AC agrees with the reviewers that the performance improvements brought out by the proposed method are commendable and thus recommends accepting the paper. Authors should incorporate the reviewers feedback in the camera-ready. AC also finds many issues  remaining in the paper listed below, which the authors need to fix.
1. Authors need to state clearly the PDE that they are solving, explicitly stating the consistency criteria and assumptions, e.g., are the trajectories continuous? Occlusions or introduction of new objects in the scene are avoided, etc.
2.  Authors should provide ablation studies on the use of the various losses used, e.g., forward prediction and cycle consistency losses.
3. There appears to be errors in the mathematical details. For example, Eq. (1) appears to have mistakes and Eq. 3 has formatting issues. These need to be fixed as well.
4. Further, there are many neural PDE formulations proposed for predicting 2D optical flow between images. AC thinks such formulations can be extended to the setting presented in this paper (such as [a, b] below) and thus the paper should clearly provide rationale on how the proposed method is conceptually different and superior to such prior methods.

[a] Zhuang, Weihao, et al. "Optical flow regularization of implicit neural representations for video frame interpolation." APSIPA Transactions on Signal and Information Processing 12.1 (2023): e39.
[b] Cho, Seokju, et al. "FlowTrack: Revisiting Optical Flow for Long-Range Dense Tracking." Proceedings of the IEEE/CVF Conference on Computer Vision and Pattern Recognition. 2024.

---

### Decision · Program_Chairs · 2025-01-22

Accept (Poster)